# Hepatitis B Surface Antibody (Anti-HBs) Kinetics during Rituximab Chemotherapy and Performance of Hepatitis B Vaccine before Immunosuppression: Two Prospective Studies

**DOI:** 10.3390/v14081780

**Published:** 2022-08-15

**Authors:** João Marcello de Araujo-Neto, Gabriela Sousa Guimarães, Flavia Ferreira Fernandes, Marcelo A. Soares

**Affiliations:** 1Departamento de Clínica Médica, Faculdade de Medicina, Universidade Federal do Rio de Janeiro, Rio de Janeiro 21941-902, Brazil; 2Instituto Nacional do Câncer Jose Alencar Gomes da Silva, Rio de Janeiro 20230-130, Brazil; 3Hospital Federal de Bonsucesso, Rio de Janeiro 21041-020, Brazil

**Keywords:** hepatitis B, HBV reactivation, anti-HBV titer, immunosuppression, rituximab

## Abstract

Rituximab promotes strong immunosuppression leading to a high risk of hepatitis B reactivation (HBV-R) and chronic infection. Current recommendations on HBV-R prevention are expensive and poorly individualized. In resolved hepatitis B patients, previous studies suggest that anti-HBs titers before immunosuppression can predict HBV-R risk. However, guidelines claim that additional data are necessary before recommending spare drug prophylaxis in patients with high anti-HBs titers. On the other hand, in patients with no previous contact with HBV, guidelines recommend vaccine before immunosuppression despite minimal evidence available. To shed light on these knowledge gaps, two prospective studies were conducted to evaluate anti-HBs in hematological cancer patients treated with rituximab. In the first study, anti-HBs-positive patients were referred for following up antibody titers before and during immunosuppression. Patients with anti-HBs ≥ 100 mIU/mL before immunosuppression had no negative seroconversion (anti-HBs loss), in contrast to 18% of those with anti-HBs < 100 mIU/mL. In the second study, patients with no previous contact with HBV were invited to receive HBV vaccine before rituximab chemotherapy. None seroconverted with anti-HBs. In conclusion, both studies reinforce the need to review concepts about HBV prevention during immunosuppression on current guidelines. Narrowing the use of drug prophylaxis and improving vaccine indications are recommended.

## 1. Introduction

The reactivation of hepatitis B virus (HBV) during immunosuppression is a challenge to be overcome by modern medicine. Potent immunosuppressant drugs are effective for the management of pathologies such as autoimmune diseases and cancer. In contrast, immunosuppression is accompanied by the risk of HBV reactivation [1]. Viral reactivation initially presents asymptomatically and is detectable only through laboratory tests. However, if it is left unmanaged, viral reactivation can progress to a clinical phase and culminate with liver failure and death [2,3,4].

HBV reactivation during immunosuppression can occur in patients with chronic hepatitis B (anti-HBc-positive and HBsAg positive) or resolved infection (anti-HBc-positive and HBsAg negative). Broad rates of HBV reactivation have been reported previously, reaching as high as 50% of the cohort analyzed [5]. In all scenarios, the use of anti-CD20 monoclonal antibodies, such as rituximab, confers a high risk (>10%) for viral reactivation [1]. Thus, current guidelines recommend prolonged use of antivirals in patients with a high risk of HBV reactivation. However, this is an expensive strategy with insufficient rates of adherence.

In the past decade, increasing attention has been paid to anti-HBs titers before immunosuppression in patients with resolved infection. Patients with negative or low titers (<100 mIU/mL) of anti-HBs before the start of immunosuppression have a higher risk of HBV reactivation [6,7,8,9,10,11]. In a cohort study by Seto et al. involving 260 patients with lymphoma treated with rituximab, negative anti-HBs before immunosuppression was the only variable correlated independently with reactivation (hazard ratio [HR] 3.51; confidence interval [CI] 95% 1.37–9.89; *p* = 0.009) [10]. Conversely, it seems that patients with anti-HBs titers ≥ 100 mIU/mL have a much lower risk of HBV reactivation during immunosuppression with rituximab for lymphoma [6,7]. In another cohort study of patients with lymphoma treated with rituximab, there was no reactivation in the subgroup of patients with anti-HBs ≥ 100 mIU/mL before rituximab and 8 reactivations in the subgroup with anti-HBs < 100 mIU/mL (0% vs. 14%; *p* = 0.007) [6].

It is assumed that negative seroconversion (loss of anti-HBs) is an event that precedes HBV reactivation. Thus, many experts advocate that anti-HBs titers before immunosuppression should guide the prevention strategy for HBV reactivation [5,6,12]. This could reduce the indication of antiviral prophylaxis for HBV reactivation, restricting its indication for patients with anti-HBS < 100 mIU/mL before immunosuppression. Despite this growing evidence, there is still no specific recommendation on guidelines to customize prevention of reactivation based on anti-HBs titers before chemotherapy.

The knowledge of the risk of reactivation based on the dichotomous anti-HBs titer (higher or lower than 100 mIU/mL) before immunosuppression seems insufficient to make further advances with respect to guideline improvement. Therefore, we think that it is also necessary to understand the kinetics of anti-HBs during immunosuppression. It seems that the loss of anti-HBs during immunosuppression is more common in patients with anti-HBs pre-treatment titers below 100 mIU/mL [9]. There are only a small number of studies dedicated to understanding anti-HBs kinetics during immunosuppression, and most without rituximab use [9,13,14]. It is also possible that, in the future, decreasing anti-HBs titers may be the trigger for starting pharmacological strategies for preventing reactivation, thus sparing antivirals in those patients who maintain higher anti-HBs titers during immunosuppression.

Looking beyond, strategies to raise anti-HBs titers before chemotherapy can also be useful for reducing antiviral prophylaxis during immunosuppression. Unfortunately, HBV vaccine performance in patients receiving rituximab is largely unknown. In 2018, Hocker et al. demonstrated that HBV vaccination before immunosuppression for kidney transplantation could elevate anti-HBs titers and lower the risk of losing the antibody after immunosuppression (odds ratio [OR] 0.47; *p* = 0.022) [14].

The EASL (European Association for The Study of The Liver) practice guidelines suggest that patients without previous contact with HBV and who are candidates for immunosuppression should be vaccinated [15]. The American College of Rheumatology also recommends HBV vaccination before immunosuppression [16]. However, there are still many open questions regarding the time of vaccination and dosing schedule in the context of immunosuppression [17]. It is known that patients using anti-CD20 drugs have an unsatisfactory response to several other vaccines compared with healthy individuals [18,19,20,21,22]. Therefore, it remains unknown how HBV vaccine should be used in rituximab-immunosuppressed patients.

To address these issues, we conducted two prospective studies in patients with hematological cancer treated with rituximab-containing chemotherapy. The subgroup of patients with positive anti-HBs before immunosuppression was forwarded to the first study, which evaluated anti-HBs kinetics during rituximab use. The subgroup of patients with negative anti-HBs before immunosuppression was enrolled in the second study in which response to HBV vaccine was assessed.

## 2. Materials and Methods

### 2.1. Patients and Study Design

There were two studies:Study 1 (anti-HBs kinetics): observational with both prospective and retrospective data collection.Study 2 (vaccine): interventional with prospective data collection.

Both experiments were conducted at the José Alencar Gomes da Silva National Cancer Institute (INCA), Brazil. This center is a tertiary referral unit for cancer treatment in Rio de Janeiro. The studies were approved by the hospital ethics committee. Patients voluntarily accepted to participate and signed an informed consent. From May 2018 to May 2019, all consecutive adult patients (>18 years old) with indication for the use of rituximab to treat cancer were invited to participate in the studies. On the retrospective arm, all medical records from patients who received rituximab at INCA from January 2014 to May 2018 were reviewed. In both study, data related to hematological disease, sociodemographics and treatment were stored in anonymized databases.

Study 1:

From May 2018 to May 2019, patients with serology compatible with HBsAg-negative/anti-HBs-positive, independent of anti-HBc status, were referred to study 1. The aim of the study was following up anti-HBs titers before and after initiating rituximab-containing chemotherapy. One measurement of anti-HBs was performed between 1 and 90 days before the start of immunosuppression with rituximab. A second anti-HBs measurement was collected three to twelve months after initiating rituximab-containing chemotherapy. We also collected the retrospective data from January 2014 to May 2018 of patients who had positive titers of anti-HBs as measured by the hospital’s laboratory 1 day to 90 days before initiating rituximab and also another measurement three to twelve months after the first dose of rituximab.

2.Study 2:

Patients with serology compatible with anti-HBc-negative/HBsAg-negative/anti-HBs-negative were invited to receive hepatitis B vaccine before initiating chemotherapy. Patients in the vaccine arm of the study were forwarded to receive vaccination (3 doses) at 0, 1, and 6 months or 0, 7, and 21 days. Anti-HBs was first measured 3 months after the first dose of the vaccine, and a second anti-HBs analysis was performed at least 1 month after the last dose of vaccination on the patients at 0-, 1-, and 6-months schedule. Recombivax-HB^®^ (Merck & Co., Rahway, NJ, USA) and Engerix-B^®^ (GlaxoSmithKline Biologicals, Rixensart, Belgium) vaccines were accepted for this research. Both were administered at the usual doses of 1 mL (20 mcg) and applied intramuscularly to the right deltoid, following good clinical practices.

HBsAg-positive and/or HBV-DNA-positive patients were excluded from both studies. Patients who were on hemodialysis or HIV-positive were also excluded from both studies.

### 2.2. Laboratory Measurements

Serum anti-HBs was measured according to the manufacturer’s instructions (LIAISON^®^ XL murex anti-HBs; Diasorin S.p.A.^®^, Saluggia, Italy). The lower limit of detection of serum anti-HBs was 10 mIU/mL, and the upper limit was 1000 mIU/mL. Results ≥ 10 mIU/mL were considered to be positive.

An anti-HBs drop rate was calculated using the formula:
[1 − (anti-HBS after immunosuppression/anti-HBs before immunosuppression)] × 100(1)

### 2.3. Statistical Analyses

Statistical analyses were performed using SPSS version 24.0 (SPSS, Chicago, IL, USA). Continuous variables were expressed as median (standard deviation). Categorical variables were analyzed based on the frequency. Chi-square or Fisher’s exact tests were used to compare categorical variables. For comparative analysis of continuous variables, Student’s *t*-test or Mann–Whitney *U* test was used, depending on whether the distribution was parametric or non-parametric. A two-sided *p*-value of <0.05 was considered statistically significant.

## 3. Results

### 3.1. Anti-HBs Kinetics Study

During the study period, 235 patients were prescribed rituximab-containing chemotherapy, of whom 38 (16.1%) were HBsAg negative/anti-HBs-positive. Thirteen patients who died before collecting the second anti-HBs, were lost to follow-up, or did not have appropriate laboratory results were excluded from the study (Figure 1).

The median age of the patients was 65 years (range, 21–86 years) and the predominant sex was male (64%). The most common indication for rituximab prescription was lymphoma (92%). Because of the advanced age of the participants, the most common etiology of anti-HBs was resolved infection (80%). The frequency of medications associated with rituximab is detailed in Table 1.

Considering all the study arm participants, the median anti-HBs before and after rituximab administration were 162 (10.4–1000) mIU/mL and 83 (3–1000) mIU/mL, respectively (*p* = 0.04) (Table 2).

The mean time between the measurements of anti-HBs (before and after rituximab) was 5.5 (±2.5) months, and the mean amount of rituximab doses was 4.4 (±1.9).

The median anti-HBs before chemotherapy in the subgroup of patients with anti-HBs ≥ 100 mIU/mL was 444.5 (131–1000) mIU/mL (Table 3). Only two (14.2%) patients had anti-HBs ≥ 100 mIU/mL before chemotherapy which dropped to less than 100 mIU/mL after immunosuppression (Figure 2). The anti-HBs before chemotherapy in these patients were 131 and 236 mIU/mL, respectively. After immunosuppression, their titers were 74.8 mIU/mL and 50.7 mIU/mL, respectively. No patient had negative seroconversion (loss of anti-HBs).

The median anti-HBs before chemotherapy in the subgroup of patients with anti-HBs < 100 mIU/mL was 29.2 (10–66) mIU/mL (Table 3). In this subgroup, two (18%) patients had negative seroconversion (anti-HBs < 10 mIU/mL). Both had low titers of anti-HBs, 11.4 mIU/mL and 29.3 mIU/mL, respectively, before immunosuppression (Figure 2).

Considering all patients, the median anti-HBs drop rate was 26.1% (−110% to 87%). The subgroups of anti-HBs categorized as <100 mIU/mL and ≥100 mIU/mL before immunosuppression showed median anti-HBs drop rates of −13.7% and 33%, respectively (*p* = 0.25). The time between anti-HBs dosages and the amount of rituximab doses did not significantly affect the anti-HBs drop rate.

Although the median anti-HBs values before and after immunosuppression were lower in the subgroup of patients with resolved infection (anti-HBc-positive/anti-HBs-positive) than in the vaccine (anti-HBc-negative/anti-HBs-positive) subgroup, there was no statistical difference between the groups (Table 4). The drop rate was lower in patients with resolved infection (0.05 vs. 73.68%; *p* = 0.01).

### 3.2. Vaccine Study

During the study period, 58 patients were prescribed rituximab-containing chemotherapy, of whom 34 (58.6%) were HBsAg-negative/anti-HBs-negative/anti-HBc-negative. Of these, 18 patients were included in the study. Three patients died from complications associated with hematological malignancy before anti-HBs analysis after vaccination (Figure 3).

The median age of the participants was 67 years (range 25–78 years). The male sex was predominant (55.2%). Non-Hodgkins lymphoma (83.3%) was the most common cause of hematological malignancy requiring immunosuppression with rituximab. The majority of patients (77.8%) received a vaccine schedule of 0, 7, and 21 days. Recombivax-HB was administered to 12 (66.7%) patients (Table 5).

The median time between the dosage of HBV serology and the administration of the first dose of rituximab was 14.5 days (0–2104). The majority of the patients (88.9%) received the first dose of vaccine at a maximum of 3 days before the first dose of chemotherapy.

No patients developed anti-HBs seroconversion after vaccination.

## 4. Discussion

The anti-HBs kinetics study provided interesting new data on the field. As mentioned before, it is well documented that patients with anti-HBs titers > 100 mIU/mL have a lower risk of HBV reactivation during immunosuppression [6,7,8,9,10,11]. However, little is known about the dynamics of anti-HBs titers during immunosuppression.

Pei et al. studied anti-HBs titers in 29 patients with B-cell lymphoma before and after the administration of rituximab [9]. It was shown that anti-HBs titers after chemotherapy were significantly lower when compared to before immunosuppression (1.82 log mIU/mL ± 0.11 vs. 1.48 log mIU/mL ± 0.14; *p* < 0.001). Our cohort also demonstrated lower medians of anti-HBs titers after chemotherapy with rituximab (162 mIU/mL vs. 83 mIU/mL; *p* = 0.04).

In the study by Pei et al., eight (42%) of 19 patients became negative for anti-HBs after immunosuppression [9]. In our cohort, this phenomenon occurred in only two (8%) out of 25 patients, and both had anti-HBs titers before immunosuppression below 100 mIU/mL. Moreover, among the patients with baseline anti-HBs titers ≥ 100 mIU/mL, only two patients had a reduction below 100 mIU/mL after immunosuppression and none developed negative seroconversion.

Based on our data, we hypothesize that patients with anti-HBs ≥ 100 mIU/mL before immunosuppression have a much lower risk of HBV reactivation because the maintenance of elevated anti-HBs titers during chemotherapy may inhibit HBV covalently closed circular DNA (ccc-DNA) transcription permanently, blocking the first event in the viral reactivation cascade.

A well-defined sequence of events occurs in HBV reactivation: emergence of HBV-DNA, appearance of serum of HbsAg, increase in aminotransferases, and liver dysfunction [5,23]. With our data, we suggest that a new step should be added to this cascade of viral reactivation events. The loss of anti-HBs should precede the classic first event, HBV-DNA emergence.

This paves the way for a new strategy for preventing HBV reactivation in patients with resolved infection: monitoring anti-HBs titer and only starting drug prophylaxis if anti-HBs became lower than 100 mIU/mL. This can save costs and reduce polypharmacy in patients undergoing multiple chemotherapies. In our cohort of Western patients, 12.8% of the patients who underwent rituximab chemotherapy presented with resolved infection and 8.5% had anti-HBs ≥ 100 mIU/mL before immunosuppression. In localities with a high prevalence of HBV infection, the impact of this new approach can have even a greater magnitude.

Recently, Tsou et al. published a study on the costs of drug prophylaxis for HBV reactivation in patients with lymphoma undergoing rituximab in Taiwan [24]. The estimated cost of diagnosing and treating patients with resolved infection was USD 1922 to USD 7864 per patient. In this study, the cost was estimated using incremental cost-effectiveness ratio to prevent HBV-related death or liver decompensation. This value was estimated at USD 98,745 to USD 155,355, depending on the duration of prophylaxis after the end of the last dose of rituximab (6 vs. 12 months). However, the author points out that if prophylaxis was applied only to anti-HBc-positive/anti-HBs-negative patients, this cost could fall by 30% to 40% [24]. Although the strategy of drug prophylaxis is enshrined in the medical literature, it needs a critical look, especially in the context of countries with limited health resources and high HBV prevalence.

Another interesting finding of this study is that the median anti-HBs titers before and after immunosuppression was higher in the subgroup of vaccinated patients. However, the drop rate of anti-HBs was lower in patients with anti-HBs from resolved infection than in those vaccinated (0.05% vs. 73.68%; *p* = 0.01). There was no correlation between the anti-HBs drop rate and the number of doses of rituximab or time of immunosuppression. The number of doses of rituximab and the time of exposure to immunosuppression in our sample population is also similar to those in the few relevant studies in the available literature [6,9,10]. This suggests the existence of individual immunological mechanisms that allow different drop rates in anti-HBs during immunosuppression. Anti-HBs ≥ 100 mIU/mL before immunosuppression can be an indirect way to identify patients with favorable immunological features not to reactivate HBV.

Despite the fact that medical societies formally recommend hepatitis B vaccination for patients who are candidates for immunosuppression, literature on vaccine performance in this scenario is scarce [15,16]. Furthermore, there are no guidelines to standardize this procedure.

It is well known that hepatitis B vaccines have an inferior performance in patients with advanced age and impaired immunity [18,19,20,21,22]. In 2018, Intongkan et al. published data on HBV vaccine from a cohort study of 8 patients using rituximab for rheumatologic diseases with a successful immunization rate of 25% [25].

Our data refer to a larger cohort (n = 18); however, none of the patients responded to the vaccine (0%). It is important to highlight that our study population had unfavorable characteristics for immunization, such as advanced age, active hematological disease, use of highly immunosuppressive chemotherapy (rituximab) in high doses, and combination with other immunosuppressants such as vincristine, cyclophosphamide, doxorubicin, and corticosteroids. Nevertheless, it is notable that most of the patients (92.8%) in our cohort received the first dose of the vaccine at very short time intervals before initiating rituximab-containing chemotherapy (≤3 days).

We believe that these data should not completely preclude the use of HBV vaccine before immunosuppression with rituximab. HBV recombinant DNA vaccine is safe in immunosuppressed patients and has a low cost [26]. Therefore, new studies with different designs should be encouraged in the oncological population. In our cohort, the median time between HBV serology and the first dose of chemotherapy was 14.5 (0–2104) days. In a cohort of cancer patients from the Mayo Clinic (Rochester, MN, USA), the median time between HBV screening serology and the start of chemotherapy was 6 months [27]. Therefore, our results may be superseded with a new study design involving the identification of patients who were candidates for the vaccine earlier and maybe their vaccination with an accelerated scheme before initiating rituximab therapy.

Our study has some limitations, mainly the size of the cohort and the suboptimal HBV vaccination strategy. Therefore, we think that it must be validated in other centers.

## 5. Conclusions

Our research group thinks that a possible strategy for preventing HBV reactivation in anti-HBc-positive/HBsAg-negative patients can be the administration of the hepatitis B vaccine in booster doses before initiating rituximab. It is biologically plausible to conclude that the increase in anti-HB titers may have a protective effect against viral reactivation and it should be evaluated in future studies.

## Figures and Tables

**Figure 1 viruses-14-01780-f001:**
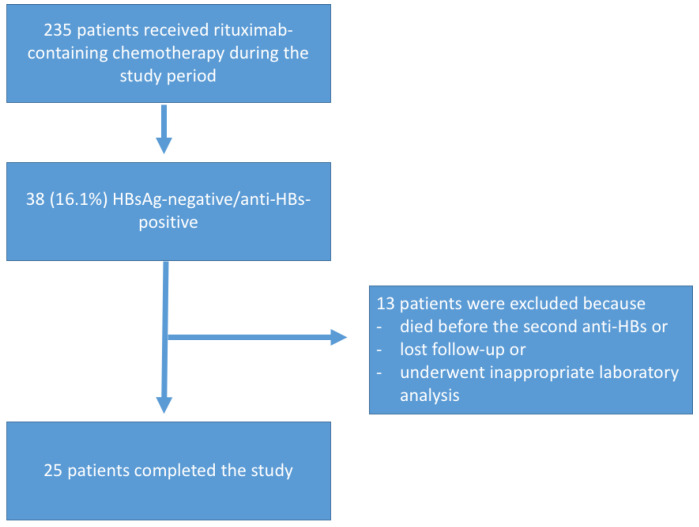
Flowchart of subjects who participated in the anti-HBs kinetics group.

**Figure 2 viruses-14-01780-f002:**
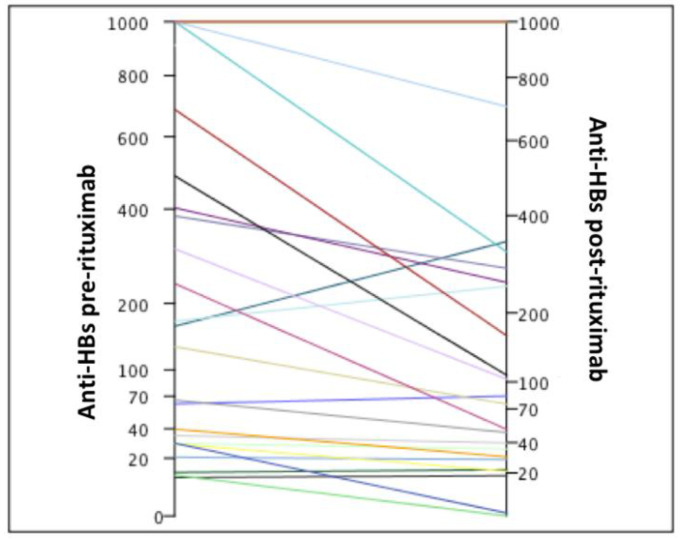
Individual values of anti-HBs before and after immunosuppression.

**Figure 3 viruses-14-01780-f003:**
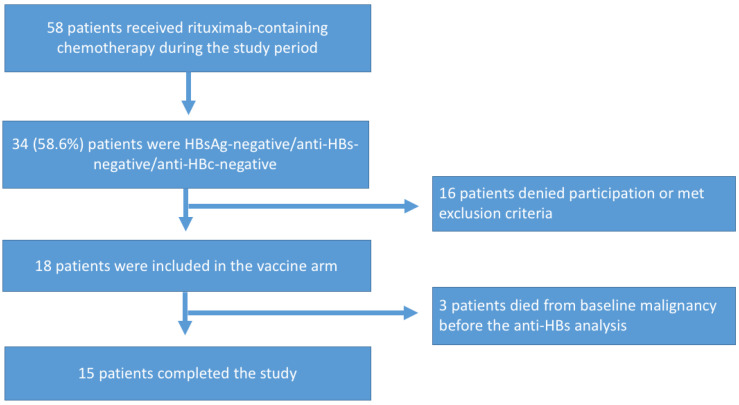
Flowchart of subjects included in the vaccine study.

**Table 1 viruses-14-01780-t001:** Baseline characteristics of anti-HBs kinetics group.

Characteristics	N = 25
**Age in years (minimum–maximum)**	65 (21–86)
**Sex**	
Male (%)	16 (64%)
Female (%)	9 (36%)
**Hematologic malignancy**	
Non-Hodgkin lymphoma (%)	23 (92%)
Leukemia (%)	2 (8%)
**Anti-HBS origin**	
Resolved infection (anti-HBc-positive) (%)	20 (80%)
Vaccine (anti-HBc-negative) (%)	5 (20%)
**Immunosuppressants used with rituximab**	
Vincristine (%)	20 (80%)
Cyclophosphamide (%)	17 (68%)
Doxorubicin (%)	14 (56%)
Hydrocortisone (%)	8 (32%)
Etoposide (%)	3 (12%)

**Table 2 viruses-14-01780-t002:** Serological profile of participants and rituximab exposure.

Characteristics	N = 25	*p*-Value
**Anti-HBs (mIU/mL)**		
Before rituximab	162 (10.4–1000)	0.04
After rituximab	83 (3–1000)	
**Anti-HBs before rituximab (n)**		
Anti-HBs ≥ 100 mIU/mL (%)	14 (56.0%)	
Anti-HBs < 100 mIU/mL (%)	11 (44.0%)	
**Anti-HBs after rituximab (n)**		
Anti-HBs ≥ 100 mIU/mL (%)	12 (48.0%)	
Anti-HBs < 100 mIU/mL (%)	13 (52.0%)	
**Time (months) between the first and second anti-HBs measurements.**	5.5 (±2.5)	
**Amount of rituximab doses between first and second anti-HBs measurements.**	4.4 (±1.9)	

**Table 3 viruses-14-01780-t003:** Median anti-HBs titers before and after rituximab therapy categorized by anti-HBs before immunosuppression.

	Anti-HBs before Immunosuppression (Minimum–Maximum)	Anti-HBs after Immunosuppression (Minimum–Maximum)	*p*-Value
**Anti-HBs before rituximab < 100 mIU/mL**	29 (10–66)	28 (3–83)	<0.001
**Anti-HBs before rituximab ≥ 100 mIU/mL**	444 (131–1000)	269 (51–1000)	

**Table 4 viruses-14-01780-t004:** Comparison of anti-HBs titers and drop rate categorized by the etiology of anti-HBs.

	Anti-HBc-Positive Anti-HBs-Positive (Resolved Infection) (n = 20)	Anti-HBc-Negative Anti-HBs-Positive (Vaccine) (n = 5)	*p*-Value
**Anti-HBs before rituximab** **(minimum-maximum)**	98 (10.4–1000)	307 (11.4–486)	0.14
**Anti-HBs after rituximab** **(minimum-maximum)**	79 (3.7–1000)	103 (3–255)	0.31
**Anti-HBs drop rate**	0.05% (−110–87)	73.68% (37–79)	0.01

**Table 5 viruses-14-01780-t005:** Baseline characteristics of the patients in the vaccine arm.

Characteristics	N = 18
**Age (years)**	
Median (minimum–maximum)	67 (25–78)
Sex	
Male (%)	10 (55.2%)
Female (%)	8 (44.4%)
**Hematologic malignancy**	
Non-Hodgkin lymphoma (%)	15 (83.3%)
Hodgkin lymphoma (%)	1 (5.6%)
Leukemia (%)	2 (11.1%)
**Vaccine**	
Recombivax-HB^®^ (%)	12 (66.7%)
Engerix-HB^®^ (%)	5 (27.7%)
Recombivax-HB + Engerix-HB^®^ (%)	1 (5.6%)
**Vaccination schedule**	
0/1/6 months (%)	4 (22.2%)
0/7/21 days (%)	14 (77.8%)
**Time between baseline HBV serology and first dose of rituximab**	
Median (minimum–maximum)	14.5 (0–2104) days
**Time between vaccine and first dose of rituximab**	
>3 days (%)	2 (11.1%)
≤3 days (%)	16 (88.9%)
**Anti-HBs seroconversion after vaccine**	
n (%)	0 (0%)

## Data Availability

Not applicable.

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
