# Peer review of "Hepatitis B Surface Antibody (Anti-HBs) Kinetics during Rituximab Chemotherapy and Performance of Hepatitis B Vaccine before Immunosuppression: Two Prospective Studies"

_viruses, 2022, doi:10.3390/v14081780_

Round 1

Reviewer 1 Report

I have no further recommendation

Author Response

Thank you very much for your comments.

Reviewer 2 Report

Dear Authors,

After reading the new version of your manuscript, i would like to ask to define EASL in the introduction section, page 96.

Be aware that you must use the right number of page and line

Author Response

Dear reviewer

EASL is the abbreviation for European Association for The Study of The Liver.

It can be found in line 79 of the manuscript.

Thank you very much for your review and comments.

This manuscript is a resubmission of an earlier submission. The following is a list of the peer review reports and author responses from that submission.

Round 1

Reviewer 1 Report

The aim of this manuscript is to conduct two prospective studies, in patients with hematological cancer, treated with rituximab-containing chemotherapy, in order to evaluate more conservative strategies to prevent HBV reactivation. In this context, this manuscript provides new immune insights in the field of viral hepatitis, with significant future prospecitves for the development of preventive and prophylactic therapies.

Even if the manuscript provides an organic overview, with a densely organized structure and based on well-synthetized evidence, there are aspects to be mentioned, to make the article fully readable. For these reasons, the manuscript requires minor changes.

Please find below an enumerated list of comments on my review of the manuscript:

INTRODUCTION:

LINE 30: The manuscript will benefit from providing a brief and organic description oh Hepatitis B Virus, considered as a global health problem: specifically, it will be useful to remark the most significant features of this small enveloped virus, providing recent insights on its molecular and genetic characteristics, often associated to interesting pathological outcomes (see, for reference: Glebe, D.; Goldmann, N.; Lauber, C.; Seitz, S. HBV Evolution and Genetic Variability: Impact on Prevention, Treatment and Development of Antivirals. Antiviral Research. 2021; DOI:10.1016/j.antiviral.2020.104973).

LINE 79: Hepatitis B Virus is the major cause of acute and chronic liver disease and a global health problem: to this aim, a pivotal role in the contrast of HBV infection is reserved to HBV vaccine, whose main task is to provide a persistent and long-term immunogenicity. In this context, an HBV vaccine not only prevents virus trasmission, but mainly reduces the global burden of HBV – associated disease. Although HBV vaccination has been carried out for several years, the debate on the duration of its protection is still open, specifically in the context of immunosuppression (see, for reference: Mastrodomenico, M.; Muselli, M.; Provvidenti, L.; Scatigna, M.; Bianchi, S.; Fabiani, L. Long-term immune protection against HBV: Associated factors and determinants. Hum. Vaccines Immunother. 2021).

In conclusion, this manuscript is densely presented and well organized, based on well-synthetized evidences. The authors were lucid in their style of writing, making it easy to read and understand the message, portrayed in the manuscript. Besides, the methodology design was rigorous and appropriately implemented within the study. However, many of the topics are very concisely covered. Moreover, this research have futuristic importance and could be potential for future research. However, I have minor comments only for the introductive section, for improvement before acceptance for publication. The article is accurate and provides relevant information on the topic and I suggest minor changes to be made in order to maximize its scientific impact. I would accept this manuscript, if the comments are addressed properly.

Reviewer 2 Report

This work involves two small studies that are both relevant to clinical practice, but both flawed. The main issue is the study size(s), and the fact that many of the eligible patients did not join. A second issue is that the two studies are not really related, and therefore the entire manuscript is a bit difficult to follow. 
The authors make the point that the breakpoint of 100 IU/mL is relevant for sero-reversion (not “negative seroconversion”, a somewhat confusing term), but this was not seen in any of the naturally infected patients, only in vaccinated patients. Since the ones who have received vaccine have a very low risk for being infected with HBV in the future (they are over 65 year of age  at mean), I am not sure how important this is for the field, as th new infection rate is likely low. There is no recommendation in this manuscript to boost the patients prior to rituxin therapy for those with a low starting value. It is important to remember that in the early days of hepatitis B vaccination, the CDC recommended boosting everyone if they fell below 10 IU/ml titer, but data emerged that anyone who had a achieved a protective titer of 10 IU/ML remained protected even if they fell below that level. That is, once protected, boosting is not needed, and it is not clear if it would be useful in this group, as an appropriate memory response could still be mounted. I would not necessarily boost a patient with a low value at low risk for HBV (core antibody and surface antibody negative). It is also unlikely that antivirals would be used in this group, as the authors somewhat imply.
The second study refers to the fact that people who are vaccinated close to the implementation of rituxin therapy do not respond appropriately to vaccine. What is needed is a study in which patients are vaccinated at intervals more distant from their immunosuppression, say 3, 6 and 9 months before this drug therapy, to see when this effect takes place. Since 50% of adults convert to a protective titer after two doses, 3 months is probably an adequate window. The finding in this paper of non-conversion is important, but fits poorly with the observational study critiqued above. 
Thus, the major issue with this paper is that the knowledge gained does not advance either field in terms of change in practice or real new knowledge.
Points
•    In the abstract the vaccine trial should not be in the fifth sentence, which splits the presentation of the observational study, but as a separate section after that is complete
•    Those patients who had a protective titer and lose this titer should not be referred to as seroconversion or negative seroconversion (both terms are used and this is confusing), and I would use the term “sero-reversion” and define this in the methods
•    The fact that 34% of the patients died in the observational arm, leaving only 25 to assess is problematic
•    The AUROC only uses two points and if included should show a confidence interval, which would be extreme. This analysis needs to be removed, as it is too underpowered. 
•    The use of higher dose HBV vaccine can result in higher tiers in some difficult population. This use should ne mentioned in the discussion. This work involves two small studies that are both relevant to clinical practice, but both flawed. The main issue is the study size(s), and the fact that many of the eligible patients did not join. A second issue is that the two studies are not really related, and therefore the entire manuscript is a bit difficult to follow. 
The authors make the point that the breakpoint of 100 IU/mL is relevant for sero-reversion (not “negative seroconversion”, a somewhat confusing term), but this was not seen in any of the naturally infected patients, only in vaccinated patients. Since the ones who have received vaccine have a very low risk for being infected with HBV in the future (they are over 65 year of age  at mean), I am not sure how important this is for the field, as th new infection rate is likely low. There is no recommendation in this manuscript to boost the patients prior to rituxin therapy for those with a low starting value. It is important to remember that in the early days of hepatitis B vaccination, the CDC recommended boosting everyone if they fell below 10 IU/ml titer, but data emerged that anyone who had a achieved a protective titer of 10 IU/ML remained protected even if they fell below that level. That is, once protected, boosting is not needed, and it is not clear if it would be useful in this group, as an appropriate memory response could still be mounted. I would not necessarily boost a patient with a low value at low risk for HBV (core antibody and surface antibody negative). It is also unlikely that antivirals would be used in this group, as the authors somewhat imply.
The second study refers to the fact that people who are vaccinated close to the implementation of rituxin therapy do not respond appropriately to vaccine. What is needed is a study in which patients are vaccinated at intervals more distant from their immunosuppression, say 3, 6 and 9 months before this drug therapy, to see when this effect takes place. Since 50% of adults convert to a protective titer after two doses, 3 months is probably an adequate window. The finding in this paper of non-conversion is important, but fits poorly with the observational study critiqued above. 
Thus, the major issue with this paper is that the knowledge gained does not advance either field in terms of change in practice or real new knowledge.
Points
•    In the abstract the vaccine trial should not be in the fifth sentence, which splits the presentation of the observational study, but as a separate section after that is complete
•    Those patients who had a protective titer and lose this titer should not be referred to as seroconversion or negative seroconversion (both terms are used and this is confusing), and I would use the term “sero-reversion” and define this in the methods
•    The fact that 34% of the patients died in the observational arm, leaving only 25 to assess is problematic
•    The AUROC only uses two points and if included should show a confidence interval, which would be extreme. This analysis needs to be removed, as it is too underpowered. 
•    The use of higher dose HBV vaccine can result in higher tiers in some difficult population. This use should ne mentioned in the discussion. 

Reviewer 3 Report

  • In introduction, there is a need a add about the global burden of Hepatitis b virus. You can cite the paper from Polaris published in lancet Gastroenterology in 2018.
  • There is need to add the global burden of cancers and then the lymphoma, you are studying in this paper. You may cite latest Global Burden of Disease papers on cancer.
  • This is the first kind of study, I am reviewing and reading in which results of two different studies and presented together.
  • There is need to add details about the ethical review committee, who approve this study.
  • The results of the study are very important, especially the use of HBV vaccine before immunosuppression.
  • There is need to improve the English language of the paper.
  • Our paper is well written and well presented and scientifically sound.